# *Beneath Surface Similarity:* Large Language Models Make Reasonable Scientific Analogies after Structure Abduction

**Siyu Yuan**$^\heartsuit$, **Jiangjie Chen**$^\spadesuit$*, **Xuyang Ge**$^\spadesuit$, **Yanghua Xiao**$^{\spadesuit\clubsuit}$, **Deqing Yang**$^{\heartsuit}$*

$^\heartsuit$School of Data Science, Fudan University
$^\spadesuit$Shanghai Key Laboratory of Data Science, School of Computer Science, Fudan University
$^\clubsuit$Fudan-Aishu Cognitive Intelligence Joint Research Center
syyuan21@m.fudan.edu.cn, {jjchen19,xyge20,shawyh,yangdeqing}@fudan.edu.cn

## Abstract

The vital role of analogical reasoning in human cognition allows us to grasp novel concepts by linking them with familiar ones through shared relational structures. Despite the attention previous research has given to word analogies, this work suggests that Large Language Models (LLMs) often overlook the structures that underpin these analogies, raising questions about the efficacy of word analogies as a measure of analogical reasoning skills akin to human cognition. In response to this, our paper introduces a task of analogical structure abduction, grounded in cognitive psychology, designed to abduce structures that form an analogy between two systems. In support of this task, we establish a benchmark called SCAR, containing 400 scientific analogies from 13 distinct fields, tailored for evaluating analogical reasoning with structure abduction. The empirical evidence underlines the continued challenges faced by LLMs, including ChatGPT and GPT-4, in mastering this task, signifying the need for future exploration to enhance their abilities.[1]

## 1 Introduction

Analogical reasoning is one of the foundations of human cognition, which helps humans understand complex and unfamiliar concepts by relating them to familiar ones (Gentner, 1983; Hofstadter, 2001; Hofstadter and Sander, 2013). In cognitive psychology, theories like the Structure Mapping Theory (SMT) have been proposed to explain the underlying mechanisms behind analogical reasoning (Gentner and Markman, 1997). According to SMT, individuals gain new knowledge by establishing mapping relations between familiar systems to unfamiliar systems (*system analogy*) (Bunge, 1981). As an example in Figure 1, an engineer can learn the eye cross-section by taking the analogy

---

*Corresponding authors.
[1]Resources of this paper can be found at https://github.com/siyuyuan/scar.

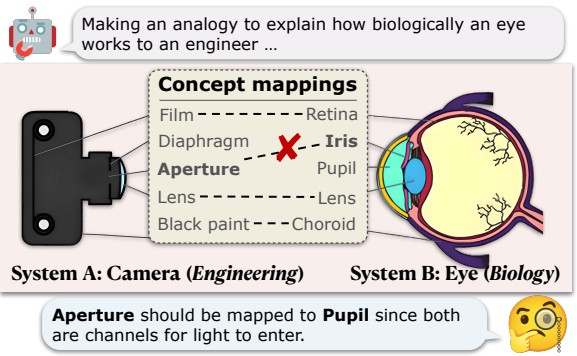

Figure 1: An example of establishing an analogy between two systems across different domains. Based on the common relational structures, an engineer can abduct *concept mappings* to learn about the cross-section of *eye* (right) with the help of *camera* structure (left).

of the camera structure since both of them exhibit common relational structures.

In this paper, we aim to evaluate the analogical reasoning ability of language models (LMs) aligning with humans. In this regard, previous work on analogical reasoning mainly focuses on word analogy (*e.g.*, "king is to man as queen is to woman"), which does not evaluate if LMs reason about the analogy between two systems in a manner akin to humans (Turney et al., 2003; Mikolov et al., 2013b; Boteanu and Chernova, 2015; Gladkova et al., 2016; Chen et al., 2022). There has been a paradigm shift in the study of analogies, moving from examining word analogies between phrases to exploring analogies between processes (Turney, 2008; Sultan and Shahaf, 2022), *e.g.*, the process of hurricanes can be analogous to the process of volcano eruptions. However, these researches remain limited to the situations within the same domains, leaving cross-domain exploration uncharted, and lack benchmarks. Large language models (LLMs, *e.g.*, Brown et al., 2020; Ouyang et al., 2022; OpenAI, 2022, 2023), despite their great abilities in

many tasks including analogy generation (Bhavya et al., 2022; Webb et al., 2022; Yuan et al., 2023), the evaluation is limited to simple word analogies. Little investigation has been done on system analogies to align with human cognition.

In this paper, we begin by evaluating and analyzing the analogical reasoning ability of LLMs on the word analogy test. Although LLMs, such as GPT-4 (OpenAI, 2023), exhibit exceptional performance in word analogy recognition, they often fail at abducing the correct structures when solving word analogies. To improve the evaluation and benchmarking of analogical reasoning for better alignment with human cognitive processes, we draw inspiration from SMT and propose an *analogical structure abduction* task. This task aims to construct the *mappings* between *concepts* in two systems based on the relational structure abduction to establish a system analogy.

For this purpose, we introduce a benchmark of **SC**ientific **A**nalogical **R**easoning with structure abduction, *i.e.*, SCAR, consisting of 400 system analogies across 13 domains with 1600 *concept mappings*, enriched with background knowledge from Wikipedia and GPT-4 generated explanations. Our experiments reveal that LLMs struggle in this task, but can be improved by incorporating background knowledge and explanations in a chain-of-thought (CoT) (Wei et al., 2022) manner.

Our contributions are summarized as follows:

- We demonstrate that word analogies do not adequately reflect the analogical reasoning ability of LMs to align with human cognition;

- We propose the analogical structure abduction task to evaluate LLMs from a cognitive perspective to align with humans;

- We develop a benchmark of scientific analogical reasoning with structure abduction, *i.e.*, SCAR, and introduce a CoT prompting method to enhance model performance on this task.

## 2   Analogical reasoning for LLMs

### 2.1   A Cognitive Perspective for Analogical Reasoning

We first introduce the cognitive foundations of analogical reasoning through the lens of Structure Mapping Theory (SMT), a psychological framework proposed by Gentner and Markman (1997).

| **I: Word Analogy Test** | |
| --- | --- |
| Query | riverbank:bridge |
| Candidates: | (A) post office:letter
*(B) floor:stairs*
(C) phone:communication
(D) train:destination |
| **II: Relational Structure Identification (RSI)** | |
| Query | riverbank:bridge::floor:stairs |
| Candidates: | (A) separate (*semantic opposite distractor*)
*(B) linked by*
(C) link (*relational opposite distractor*)
(D) adjacent (*similar distractor*) |

Table 1: Examples of word analogy and RSI task. We also give the distractor types in RSI task for a better understanding. The true answers are *highlighted*.

SMT suggests that analogy is achieved by identifying common relational structures between two systems (*i.e.*, *system analogy*) (Bunge, 1981; Gentner, 1983). Key components of SMT include

1. **Representation**: Structured systems with concepts and relations;
2. **Mapping**: Comparisons between two representations for commonalities, resulting in *structure abduction* between two systems;
3. **Evaluation**: Analogies are evaluated based on the abduced structures between the two representations.

An example of SMT is illustrated in Figure 1. The two systems, *i.e.*, camera and eye, can be represented into five concepts. Based on the relational structure, *aperture* should be mapped to *pupil* since both are channels for light to enter. Adhering to the *one-to-one mapping*, this process focuses on structure abduction to foster comprehension across both domains (Bartha, 2013).

### 2.2   *How are LLMs on word analogy test?*

Previous work adopts word analogy tests (*i.e.*, *A is to B as C is to D*) to evaluate the analogical reasoning ability of LMs. As illustrated in Table 1 (I), this task can be framed as a multiple-choice question-answering (QA) challenge. We adhere to this paradigm and test the performance of LLMs, *e.g.*, InstructGPT series (Ouyang et al., 2022), ChatGPT (OpenAI, 2022) and GPT-4 (OpenAI, 2023), on this test. We also adopt InstructGPT embeddings (text-ada-embedding-002) and follow the method proposed by Ushio et al. (2021), which converts word analogies into embeddings and select

| Model | $k$ | Word Analogy Test | | | | | | | RSI Test on E-KAR | |
| | | E-KAR | BATS | UNIT 2 | UNIT 4 | Google | SAT | Mean | Accuracy | Overlap (↑) |
|---|---|---|---|---|---|---|---|---|---|---|
| Embedding | - | 30.53 | 30.24 | 34.65 | 33.56 | 50.40 | 36.69 | 36.01 | 33.25 | 22.40 |
| InstructGPT$_{002}$ | 0 | 32.44 | 57.78 | 47.80 | 46.99 | 78.40 | 37.39 | 50.13 | 57.50 | 47.50 |
| | 1 | 38.93 | 81.90 | 50.00 | 52.78 | 91.60 | 48.96 | 60.70 | 58.50 | 49.50 |
| InstructGPT$_{003}$ | 0 | 39.31 | 82.77 | 56.14 | 58.33 | 94.40 | 47.48 | 63.07 | 61.50 | 48.50 |
| | 1 | 41.60 | 88.99 | 62.72 | 63.66 | 98.20 | 57.27 | 68.74 | 65.50 | 50.00 |
| ChatGPT | 0 | 41.22 | 81.71 | 53.07 | 52.31 | 93.80 | 49.26 | 61.90 | 64.30 | 52.47 |
| | 1 | 44.27 | 81.59 | 59.21 | 55.32 | 94.80 | 55.19 | 65.06 | 68.48 | 53.76 |
| GPT-4 | 0 | 53.05 | 92.42 | 76.32 | 71.30 | 98.80 | 74.78 | 77.78 | 71.69 | 60.47 |
| | 1 | 60.36 | **93.97** | 84.21 | **81.71** | **100.00** | **83.68** | **83.99** | 78.50 | 64.90 |
| Human | - | **77.80** | 84.85 | **87.50** | 66.66 | 99.41 | 57.00 | 78.87 | **86.43** | **98.70** |

Table 2: Accuracy on the word analogy test and RSI test with $k$-shot learning. **Overlap** indicates the ratio of correct answers on both E-KAR and RSI tests. We obtain human performance on word analogy benchmarks from the original papers. The best results are **bolded**, and the second best ones are underlined.

the answer candidate with the marginal likelihood biased perplexity.

We use six benchmarks and design instructions for LLMs to complete the tests. SAT (Turney et al., 2003), UNIT2 (Boteanu and Chernova, 2015) and UNIT4 (Boteanu and Chernova, 2015) come from educational resources. Google (Mikolov et al., 2013b) and BATS (Gladkova et al., 2016) are derived for word embedding evaluation with semantic and morphological relations. E-KAR (Chen et al., 2022) is from China's Civil Service Examinations with more complex relations and structures.[2]

The results in Table 2 show that *1)* InstructGPT embeddings perform poorly on the word analogy test; *2)* GPT-4 achieves human-level performance and providing examples improves model performance; *3)* Despite GPT-4 exceeding human performance on most word analogy benchmarks, it lags considerably behind humans on E-KAR.

However, analogical reasoning relies on identifying shared relational structures between two systems, but the word analogy test does not explicitly evaluate structural abduction. Thus, the word analogy test may not reflect model performance in analogical reasoning aligned with humans.

### 2.3 *Is the word analogy test aligned with humans?*

To confirm our hypothesis, we explore the discerning relations between word analogies for LLMs. We analyze word analogies in E-KAR to explore whether LLMs can establish structural relations.

As shown in Table 1 (II), we define a relational structure identification (RSI) test where

LLMs select the relation constituting the analogy from four options. We construct a benchmark using 700 E-KAR test data, where annotators identify the correct relation of the word analogy. For distractors, annotators write the relation opposite to the golden relation (*relational opposite distractor*) and the semantic opposite relation (*semantic opposite distractor*). Besides, we convert golden and Wikidata relations (Vrandečić and Krötzsch, 2014) into InstructGPT embeddings (text-ada-embedding-002) and calculate cosine similarity. Then, annotators select an incorrect relation with the closest semantics as a distractor (*similar distractor*). Two annotators annotate each data at first, and then we hire a third annotator to select a better one as the distractor. For human performance, we test two undergraduates on the RSI test with their results averaged.

We evaluate LLMs on the RSI task and calculate the **Accuracy** and **Overlap**. Overlap is a metric that represents the ratio of data samples that are correctly identified in both tasks to the total number of samples correctly identified in at least one of the tasks. A higher overlap suggests LMs tend to understand word analogies based on structure.

Table 2 shows superior model performance in the RSI test. However, the low overlap reveals LLMs doing well in the RSI test may not necessarily succeed in the word analogy test. According to SMT, analogical reasoning is based on identifying common relational structures between two systems. Such discrepancy indicates that the word analogy test is not aligned with humans, and we need a new way of evaluating and benchmarking analogical reasoning to align with humans.

---

[2]Details on the benchmarks and prompt templates are provided in Appendix B.

# 3 SCAR: Scientific Analogical Reasoning with Structure Abduction

## 3.1 Schema for Analogies in SCAR

In this paper, we aim to explore the analogical reasoning ability of LLMs to align with humans. Inspired by SMT, we focus on the *structure abduction* of *system analogy* (Bunge, 1981). As shown in Figure 1, two *systems* are analogous based on their common relational structure. To construct the system analogy, *concepts* in System A (*e.g.*, *Camera*) can be mapped into corresponding concepts in System B (*e.g.*, *Eye*), forming multiple one-to-one *concept mappings* (*e.g.*, *Aperture maps to Pupil*). This process facilitates analogical reasoning and enables a deeper understanding of both systems.

## 3.2 Data Collection

**System Analogy Selection**   Given that analogical reasoning is usually used in scientific problem-solving, we construct a benchmark of **SC**ientific **A**nalogical **R**easoning with structure abduction, *i.e.*, SCAR. We recruit a team of five undergraduate students with different academic backgrounds to serve as annotators for this benchmark.

The annotators are provided with guidelines of SMT to learn about identifying potential analogies based on the relational structures of systems. To assist our annotators, we furnish them with scientific analogies sourced through online research.[3] These resources contain various scientific analogies with detailed information and thus can prompt annotators to create system analogies with concept mappings. Overall, annotators manually curate 400 system analogies and define mappings between concepts based on their domain-specific expertise. We also ask the annotators to mark the domains in each system analogy. Then, we remove duplicates to collate the benchmark and conduct a review process to verify the correctness and plausibility of the analogies in the benchmark.

**Background Knowledge Retrieval**   We incorporate background knowledge into each system to facilitate the understanding of LMs and streamline the mapping process. To achieve this, we first extract the encyclopedia abstracts from Wikipedia[4] for each system. Considering that abstracts may not include all relevant concepts and could be too lengthy, we use ChatGPT (OpenAI, 2022)

| Statistic | Number |
|---|---|
| Total system analogies | 400 |
| Systems | 632 |
| Mappings | 1615 |
| Concepts in analogies | 3230 |
| Domain classes | 13 |
| Word Analogy | 3159 |
| Different mappings | 1555 |
| Different concepts | 2046 |
| Average mappings in analogies | 4.04 |
| Average concept length | 1.36 |
| Average background length | 148.81 |
| Average explanation length | 44.80 |

Table 3: Main statistics of SCAR.

to rewrite each abstract as the background. We prompt ChatGPT with human-written instructions to ensure that each revised background is limited to 500 words and encompasses all concepts in each system.[5]

**Explanation Generation**   As shown in Figure 1, *Film* maps *Retina* since both capture light and translate it into recognizable information. To rationalize analogical reasoning, we design prompts for GPT-4 to generate explanations for each concept mapping.[6] To ensure the quality of explanations, we employ two annotators to evaluate the accuracy of each explanation in SCAR. The annotation results indicate that 69.35% of the concept mapping explanations are accurate, with Fleiss's $\kappa = 0.93$ (Fleiss et al., 1981). Then we ask the annotators who created the dataset to revise the wrong explanations (495 in total) with their expertise, thereby guaranteeing the quality of the explanations.

**Bilinguality: English and Chinese**   To broaden the scope of this work, we also develop a Chinese version of SCAR through translation. We employ three native Chinese-speaking annotators to refine the machine translation provided by Google. Finally, we have a bilingual SCAR benchmark.[7]

## 3.3 SCAR Analysis

Table 3 shows the main statistics of SCAR. SCAR contains a total of 400 system analogies, with 632 systems and 1,614 concept mappings, indicating a

---

[3] The online resources are shown in Appendix C.1.
[4] https://www.wikipedia.org/

[5] The instruction template to revise backgrounds is shown in Appendix C.3.
[6] The instruction template to generate explanations is shown in Appendix C.4.
[7] The data examples of SCAR are provided in Appendix C.5.

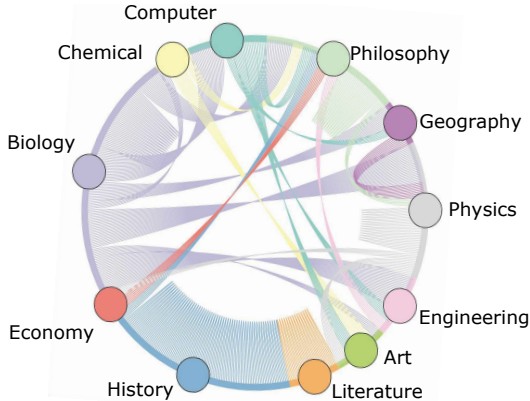

Figure 2: Domain transfer in SCAR benchmark

Table 4: An instruction template for LLMs in the analogical structure abduction task. Generated texts by LLMs are *highlighted*.

rich and complex analogy structure that can potentially challenge LLMs. The benchmark spans 13 domains for evaluating the generalizability of models across various domains. In addition, the benchmark provides backgrounds and explanations for concept mappings, serving as valuable resources to rationalize reasoning.

**Comparison with Previous Benchmarks** We compare SCAR to existing analogy resources by transforming it into word analogies. As in Fugure 1, we can obtain a word analogy, *i.e.*, *Film is to Retina as Aperture is to Pupil*. Overall, as shown in Table 3, there are 3,159 word analogies in SCAR, which exhibits a larger number of word analogies than previous benchmarks.[8]

**Domain Analysis** We consider the domains of the two systems in each system analogy as a pair, *e.g.*, (Engineering, Biology), and calculate the frequency of each pair in SCAR to derive the domain transfer distribution of SCAR, as illustrated in Figure 2. The Figure highlights interdisciplinary aspects, with prominent cross-field relationships between Biology, Engineering, and Physics, emphasizing their inherent inter-connection. SCAR shows the prevalence of within-field analogies and asserts the significance of promoting interdisciplinary connections to foster collaborative advancements in knowledge acquisition.

### 3.4 Probing Task Formulation

Our task draws inspiration from SMT, which suggests that analogical reasoning involves drawing correspondences between two systems, founded on their shared relational structure. To this end, we define the *analogical structure abduction* task to explore the analogical reasoning ability of LLMs. Given two systems:

$$S_A = \{t_1^A, t_2^A, ...., t_n^A\},$$
$$S_B = \{t_1^B, t_2^B, ...., t_n^B\},$$

this task involves establish mappings between concepts $\{t_i^A\}_i$ and $\{t_j^B\}_j$ for two systems to form an analogy between $S_A$ and $S_B$. The task requires the understanding of the relational structures between concepts in both systems and creating a one-to-one mapping between them. Table 4 shows an instruction for LLMs to generate mappings. Our evaluation assesses the accuracy of concept mappings and system analogy. A system analogy is deemed correct if all concept mappings between the two systems are accurate.

## 4 Evaluation

### 4.1 Evaluation Settings

To minimize the impact of instruction design on LLMs, we create 10 different instruction templates for LLMs in this task and select the best one to evaluate model performance. Furthermore, we also explore the ability of models to use background knowledge and the CoT prompting (Kojima et al., 2022; Wei et al., 2022) with explanations in this task. The templates are shown in Table 11.

---

[8]The detailed comparison is shown in Appendix C.6

| Method | Concept Acc. | | System Acc. | | Avg Acc. |
|---|---|---|---|---|---|
| | En | Zh | En | Zh | |
| Alpaca | 4.58 | 0.00 | 1.75 | 0.00 | 1.58 |
| w/ 1-shot | 9.97 | 14.11 | 4.50 | 5.50 | 8.52 |
| Vicuna | 9.04 | 26.42 | 4.50 | 0.12 | 10.02 |
| w/ 1-shot | 40.93 | 16.00 | 21.75 | 0.00 | 19.67 |
| InstructGPT$_{001}^{curie}$ | 3.18 | 0.00 | 1.75 | 0.00 | 1.23 |
| w/ 1-shot | 8.43 | 5.43 | 2.75 | 2.00 | 4.65 |
| InstructGPT$_{002}$ | 51.18 | 37.83 | 36.18 | 25.37 | 37.64 |
| w/ 1-shot | 54.71 | 48.82 | 40.25 | 33.50 | 44.32 |
| w/ Backg. | 51.36 | 54.86 | 34.50 | 41.25 | 45.49 |
| w/ 1-shot+Backg. | 55.46 | 54.94 | 41.30 | 45.11 | 49.20 |
| InstructGPT$_{003}$ | 52.49 | 39.91 | 36.25 | 26.50 | 38.79 |
| w/ 1-shot | 55.36 | 47.70 | 40.50 | 33.00 | 44.14 |
| w/ Backg. | 54.95 | 54.76 | 37.25 | 41.75 | 47.18 |
| w/ 1-shot+Backg. | 58.63 | 53.92 | 42.60 | 43.50 | 49.66 |
| ChatGPT | 66.52 | 66.26 | 46.61 | 52.00 | 57.85 |
| w/ 1-shot | 69.99 | 70.33 | 51.25 | 56.75 | 62.08 |
| w/ Backg. | 70.78 | 71.97 | 52.50 | 61.25 | 64.13 |
| w/ 1-shot+Backg. | 72.80 | **74.03** | 57.39 | 59.25 | 65.87 |
| GPT-4 | 73.28 | 69.77 | 58.50 | 58.50 | 65.01 |
| w/ 1-shot | 71.66 | 71.96 | 59.25 | 61.75 | 66.16 |
| w/ Backg. | 75.10 | 73.11 | 63.00 | **62.74** | 68.49 |
| w/ 1-shot+Backg. | **77.17** | 72.74 | **64.00** | 62.50 | **69.10** |
| Human | 85.94 | 88.46 | 83.37 | 86.36 | 86.03 |

Table 5: Main Results of different LLMs. We compare vanilla LLMs and LLMs with one added example (w/ 1-shot) or background (w/ Backg.). The best results are **bolded** and the second best are underlined.

## 4.2 Model Choices

We choose Alpaca (Taori et al., 2023) (7B), Vicuna (Chiang et al., 2023) (7B), In-structGPT series (Ouyang et al., 2022) (*i.e.*, InstructGPT$_{001}^{curie}$ (6.7B), InstructGPT$_{002}$ ($\geq$175B) and InstructGPT$_{003}$ ($\geq$175B)), ChatGPT (OpenAI, 2022) and GPT-4 (OpenAI, 2023). Alpaca and Vicuna are fine-tuned from a 7B LLaMA model (Touvron et al., 2023) with instructions or user-shared conversations. InstructGPT series are variants of GPT-3 (Brown et al., 2020) fine-tuned on instructions using reinforcement learning with human feedback (RLHF). ChatGPT is built on InstructGPT and trained on dialogue data using RLHF. GPT-4 is the most advanced LLM so far.

## 4.3 Overall Performance

We first compare LLMs with 0-shot, 1-shot, and background knowledge (w/ Backg.). Due to the limited input length, we excluded background knowledge from Alpaca, Vicuna, and InstructGPT$_{001}^{curie}$. For human performance, we test two graduate students, one in liberal arts and the

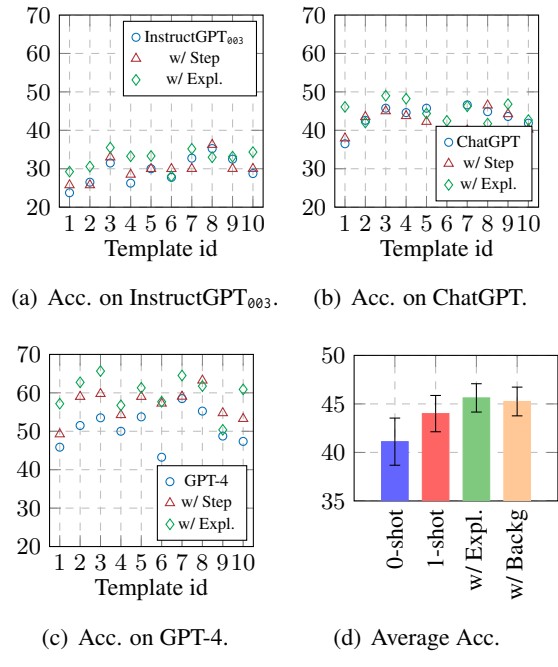

(a) Acc. on InstructGPT$_{003}$.  (b) Acc. on ChatGPT.

(c) Acc. on GPT-4.  (d) Average Acc.

Figure 3: Subfigures (a-c) show the system accuracy of LLMs enhanced with different types of CoT prompting. The average system accuracy of LLMs (*i.e.*, InstructGPT$_{003}$, ChatGPT and GPT-4) on different templates is shown in Subfigure (d). All results are from the English version of SCAR.

other in science, with their results averaged. Result in Table 5 shows that: *1)* GPT-4 achieves the best performance across both languages, and adding an example can enhance model performance. However, its ability still lags behind that of humans; *2)* Smaller models perform poorly, but training on dialogue data can improve model performance; *3)* The performance of the InstructGPT series and ChatGPT in Chinese is improved significantly by adding background knowledge, highlighting the model's struggle with domain-specific terminology in Chinese to affect their performance.

## 4.4 Analysis

**Will step-by-step reasoning help solve SCAR?** We dig into the effectiveness of the CoT prompting technique for LLMs in structure abduction. We adopt one example with the explanations of concept mappings to induce the LLMs to generate intermediate steps in analogical reasoning (w/ Expl.). One instruction template is shown in Table 11 (II). We also add "*Let's think step by step*" before each answer (w/ Step), which improves zero-shot reasoning for LLMs (Kojima et al., 2022). We conduct experiments on ten different templates to mitigate

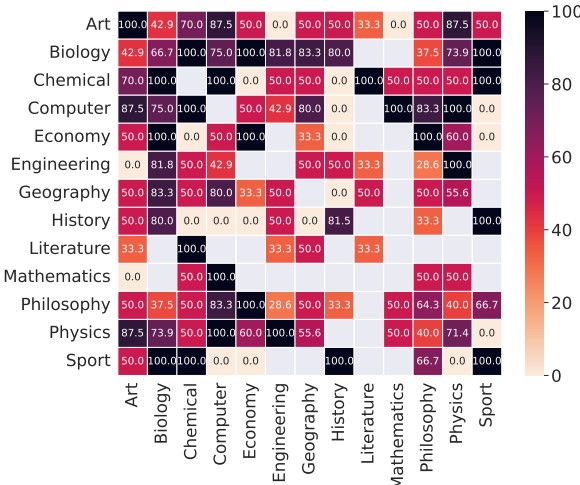

Figure 4: The heatmap depicts the system accuracy of GPT-4 across different domains.

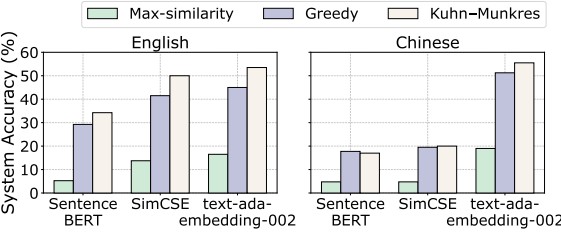

Figure 5: The system accuracy of different embedding methods (*i.e.*, Sentence-BERT (Reimers and Gurevych, 2019), SimCSE (Gao et al., 2021), `text-ada-embedding-002` (Ouyang et al., 2022)).

human design bias for LLMs. Results in Figure 3 (a-c) show that: *1)* CoT prompting enhances GPT-4 performance in structure abduction but harms ChatGPT and InstructGPT$_{003}$ performance due to flawed reasoning; *2)* CoT prompting with explanations outperforms the "*Let's think step by step*" approach, highlighting the importance of explanations in CoT prompting.

**Does instruction design affect the model performance?** To answer this question, we calculate the average system accuracy of LLMs across all templates. Results in Figure 3(d) show that LLMs are sensitive to the instruction design. However, providing an example, additional backgrounds, or utilizing CoT prompting can enhance the robustness of LLMs to the instruction design.

**Do models behave differently for analogies across various domains?** The heatmap in Figure 4 shows the system accuracy of GPT-4 across different domains.[9] We find that: *1)* The performance varies considerably for system analogies of different domains, indicating limited sensitivity of LLMs to domain boundaries; *2)* In certain domains, *e.g.*, literature, intra-domain system analogies have low accuracy, indicating LLMs may have some difficulties in these fields; *3)* System analogies between similar domains (*i.e.*, biology and chemistry) show high accuracy, demonstrating the potential for knowledge transfer.

**How do the embedding-mapping algorithms of concepts affect system analogies?** We explore creating system analogies based on the embeddings of each concept in the systems. To achieve this, we implement three distinct mapping algorithms, leveraging the cosine similarity score of embeddings to facilitate the process: *1) Max-Similarity Algorithm*: This algorithm maps each concept in System A to the concept from System B that exhibits the highest cosine similarity score, implying the same concept from System B can map to multiple concepts from System A; *2) Greedy Algorithm* (Zhang et al., 2000): This algorithm iteratively maps the concepts with the highest cosine similarity. In each iteration, the concepts with the highest similarity are mapped and *excluded from further consideration*, generating one-to-one mappings without overall optimality; *3) Kuhn–Munkres Algorithm* (Kuhn, 1955): This combinatorial optimization algorithm generates one-to-one mappings, providing a globally optimized solution.[10]

Results in Figure 5 reveal the insufficiency of the max-similarity algorithm in creating viable system analogies in SCAR. It underscores the inference that human-like analogical reasoning does not rely solely on surface-level embedding similarities. However, both the Greedy and Kuhn–Munkres algorithms display enhanced performance, suggesting that the lackluster results of the LLMs on the structure abduction task might be attributed to their weak mapping capabilities. This observation indicates that human-like reasoning could employ embeddings alongside mapping algorithms as complementary tools to deduce system analogies.

---

[9]We select the best results of GPT-4 on the English version of SCAR to draw the heatmap.

[10]Please refer to Appendix E for an in-depth explanation of the employed embedding similarity methods.

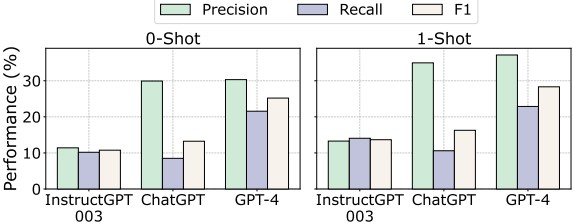

Figure 6: The performance of LLMs in the open analogical structure abduction task. All results are from the English version of SCAR.

## 4.5 Open Analogical Structure Abduction

In the above experiments, we evaluate LLMs' ability of structure abduction in a close setting, where the concepts of each system are given (as in Table 4). However, a more intriguing question arises: Can LLMs perform analogical reasoning for systems with an open set of concepts, where concepts are not explicitly given? In this case, models must first identify concepts from contexts and then generate concept mappings to form a system analogy. This *open analogical structure abduction* problem more closely simulates the process of humans discovering and acquiring knowledge.

We provide LLMs with the background description texts of systems to simulate an open setting. LLMs are expected to retrieve concepts that can be used to create mappings from backgrounds and then establish concept mappings to form system analogies.[11] For evaluation, we automatically calculate the recall of concept mappings based on SCAR to measure the correctness of newly generated mappings with annotated mappings (as in the close setting). Since some reasonable mappings may not be included in SCAR, we need to manually evaluate the precision. We randomly sample 50 data from SCAR and let LLMs generate concept mappings in the open setting. Two annotators assess the precision of the generated concept mappings with Fleiss's $\kappa = 0.86$. Then F1 score can be calculated based on precision and recall.

The results in Figure 6 show that LLMs can establish some concept mappings even when concepts are not directly given. Despite the relatively low recall, higher precision indicates LLMs' ability to form new concept mappings, which can be utilized to further improve SCAR.

---

[11]One instruction template is shown in the Appendix D.

## 5 Related Work

Analogical reasoning has been an area of interest in the AI community, primarily focusing on word analogies (Mitchell, 2021). Early researches in word analogy focus on evaluating the quality of word embeddings, which examines linear relations between words (Turney et al., 2003; Mikolov et al., 2013b; Gladkova et al., 2016) and can be effectively addressed through vector arithmetic for neural word embeddings such as Word2Vec (Mikolov et al., 2013a) and GloVe (Pennington et al., 2014).

In recent years, some studies explore analogy understanding of LMs on various benchmarks (Fournier et al., 2020; Ushio et al., 2021) and fine-tuned LMs using constructed knowledge bases of analogies to improve performance (Li et al., 2018, 2020; Yuan et al., 2023). New word analogy benchmarks with more complex relational structure (Chen et al., 2022; Czinczoll et al., 2022) and with multimodal elements (Zhang et al., 2023) (Zhang et al., 2023) are built to evaluate the performance of multilingual and multimodal models. However, a gap exists between word analogy formats and the nature of analogical reasoning in human cognition (Hofstadter, 2001; Bartha, 2013; Gentner and Maravilla, 2017), limiting the word analogy task to reflect the LLMs' analogical reasoning ability aligning with humans.

There has been a paradigm shift toward exploring analogies between situations (Turney, 2008; Sultan and Shahaf, 2022). These works are inspired by the SMT (Gentner, 1983), which aims to establish mappings between concepts in two domains based on a common relational structure. Nonetheless, Turney (2008) focuses on simple commonsense relations. Sultan and Shahaf (2022) argue that two processes with similar questioning formats can be analogies, which does not address complex structures and yield unsatisfactory performance on discovering analogies between different domains. Furthermore, some studies also explore the analogy generation of LLMs (Bhavya et al., 2022; Yuan et al., 2023; Ding et al., 2023). However, they mostly evaluate word analogies or simple analogies between two sentences, leaving complex structures in analogical reasoning unstudied. Webb et al. (2022) and Hu et al. (2023) examine the abstract language-based analogy task and evaluate the analogical reasoning ability of LLMs on this task. Compared to their task, our task requires the intensive involvement of commonsense, encyclopedic,

and cultural (e.g., idiom and historical) knowledge.

Recent researchers study AI alignment to guide AI toward achieving human preferences and ethical principles (Li et al., 2022; Rao et al., 2023; Park et al., 2023). We explore the analogical reasoning ability of LLMs with complex structure abduction, which is more aligned with human cognition.

## 6 Conclusion

In this paper, we explore the analogical reasoning ability of LLMs. We highlight word analogies neglect structures and thus can not evaluate LLMs in alignment with human cognition. To better evaluate LLMs aligning with humans, we propose an analogical structure abduction task with a new benchmark, SCAR. Experiments show that LLMs struggle with this task, but incorporating background knowledge and CoT prompting can improve their performance. We hope the SCAR can be a valuable resource to advance the research on analogical reasoning.

## Limitations

We only instruct LLMs to establish concept mappings between systems in the analogical structure abduction task, leaving the discovery of novel analogies unexplored. Such a limitation highlights the potential for future work to adopt structure abduction to uncover analogies and learn about new knowledge.

Another limitation of this work is that our evaluation of the analogical structure abduction task relies on concept mappings. Although this criterion aligns with humans, it remains a challenge for the model. Future studies can consider designing more appropriate evaluation tasks. Additionally, although we mitigated the impact of templates on model results by designing ten templates and choosing the best results for evaluation, we believe there remains room for improvement in instruction design to fully harness the capability of LLMs.

## Ethics Statement

All authors of this work abide by the provided Code of Ethics. Annotators recruited by our institution annotate the system analogies in SCAR. The annotation quality is ensured through a double-check strategy outlined in Section 3. We ensure that the privacy rights of all annotators are respected in the annotation process. As described in our paper, all annotators are compensated above the local minimum wage and consent to using the SCAR for research purposes.

## Acknowledgement

We thank the anonymous reviewers for their valuable comments, and Yikai Zhang and Shuang Li from Fudan University for their useful suggestions for the manuscript. This work is supported by the Chinese NSF Major Research Plan (No.92270121), Shanghai Science and Technology Innovation Action Plan (No.21511100401) and the Science and Technology Commission of Shanghai Municipality Grant (No. 22511105902).

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

## A Author Contributions

**Siyu Yuan** Lead the project, develop the original method and original code, lead the curation of the dataset, contribute to the original experiments, and contribute to the original manuscript.

**Jiangjie Chen** Conceptualization of the original idea, supervision of the research activity planning and execution, contribution to the original manuscript and figures, and acquisition of financial support for the project.

**Xuyang Ge** Contribute to the experiments and data curation.

**Yanghua Xiao** Provide financial support for the project and revise the manuscript.

**Deqing Yang** Provide financial support for the project, revise the manuscript, and oversee the research activity execution.

## B Word Analogy Test

### B.1 Benchmark

We compare LLMs with human performance in 6 different analogy benchmarks.

- **E-KAR** (Chen et al., 2022): A knowledge-intensive analogical reasoning benchmark from China's Civil Service Examinations (CSE), including linguistic, commonsense, encyclopedic, and cultural knowledge.

- **BATS** (Gladkova et al., 2016): This benchmark features over 1,000 analogies in four categories: lexicographic, encyclopedic, derivational, and inflectional morphology.

- **UNIT2** (Boteanu and Chernova, 2015): A benchmark using word analogy problems from an educational resource.

- **UNIT4** (Boteanu and Chernova, 2015): Similar to U2, this benchmark comes from an educational resource but is more challenging.

- **Google** (Mikolov et al., 2013b): A benchmark for intrinsic evaluation of word embeddings, containing semantic and morphological relations.

- **SAT** (Turney et al., 2003): A benchmark derived from a US college admission test with 374 word analogy problems.

## B.2 Prompt Templates for Large Language Models

As shown in Table 9, we design the instruction for LLMs to generate the answers.

## C Details of SCAR

### C.1 Resource of SCAR

To facilitate a more efficient and effective annotation process, we furnish annotators with scientific analogies sourced through online research:

- https://homework.study.com/explanation/what-is-analogy-in-science.html

- www.csun.edu/science/ref/analogy/analogy.htm

- https://science-education-research.com/teaching-science/constructivist-pedagogy/making-the-unfamiliar-familiar/science-analogies/

- www.engagefastlearning.com

| Data | # Analogy | Lang. | Backg. | Expl. |
|------|-----------|-------|--------|-------|
| SAT | 374 | En | ✗ | ✗ |
| Google | 550 | En | ✗ | ✗ |
| UNIT 2 | 252 | En | ✗ | ✗ |
| UNIT 4 | 480 | En | ✗ | ✗ |
| BATS | 1998 | En | ✗ | ✗ |
| E-KAR | 1251 | En | ✗ | ✓ |
| E-KAR | 1655 | Zh | ✗ | ✓ |
| SCAR | 3159 | En | ✓ | ✓ |
| SCAR | 3159 | Zh | ✓ | ✓ |

Table 6: Comparison between SCAR and previous analogy data source: numbers of word analogies, language (**Lang.**) and whether the benchmark has background information (**Backg.**) and explanations (**Expl.**).

For example, "*When Rutherford (following Nagoka) conceived of the atom as a miniature solar system – electrons circling the nucleus as planets circle the sun*". These resources can prompt annotators to create system analogies with concept mappings.

### C.2 Crowd-sourcing Details

We have recruited a team of five undergraduates with diverse academic backgrounds in Computer Science, History, Physics, Biology, and Chemistry. Among them, the student majoring in Computer Science has a minor in Economics, while the student majoring in History has minors in Creative Writing and Philosophy. We pay each annotator $8/h, exceeding the local minimum wage $5/h.

### C.3 Backgroud Rewriting Template

As shown in Table 10 (I), we design the instruction for ChatGPT to ensure that each revised background is limited to 500 words and encompasses all concepts relevant to each system.

### C.4 Explanation Generation Template

Table 10 (II) shows the human-written instruction for GPT-4 to generate the explanation for each mapping in SCAR.

### C.5 Data Examples of SCAR

Table 7 presents some examples of SCAR for a better understanding.

### C.6 Comparison to Previous Benchmarks

We compare SCAR with the resources related to the problem of analogy. As the existing benchmarks are based on word analogies, we transform SCAR for appropriate comparisons. We combine

| 1 | Limit Modification (Biology) | $\Longrightarrow$ | Firewall (Computer) |
|---|---|---|---|
| | Prokaryotes | $\longrightarrow$ | Computer |
| | Exogenous DNA | $\longrightarrow$ | Virus |
| | Restriction Enzyme | $\longrightarrow$ | Antivirus Software |
| | Cut Off | $\longrightarrow$ | Intercept |
| | Degradation | $\longrightarrow$ | Clear |
| 2 | Tide (Geography) | $\Longrightarrow$ | Lift (Engineering) |
| | Ocean | $\longrightarrow$ | Platform |
| | Moon | $\longrightarrow$ | Console |
| | High tide | $\longrightarrow$ | Rise |
| | Ebb and Flow | $\longrightarrow$ | Decline |
| 3 | Sound (Physics) | $\Longrightarrow$ | Light (Physics) |
| | Low | $\longrightarrow$ | Red |
| | High | $\longrightarrow$ | Violet |
| | Echoes | $\longrightarrow$ | Reflects |
| | Loud | $\longrightarrow$ | Bright |
| | Quiet | $\longrightarrow$ | Dim |
| | Horn | $\longrightarrow$ | Lens |
| 4 | Computer Systems (Computer) | $\Longrightarrow$ | Urban (Geography) |
| | Operating System | $\longrightarrow$ | Mayor |
| | Process | $\longrightarrow$ | Resident |
| | Resource manager | $\longrightarrow$ | Municipal Facilities |
| | File System | $\longrightarrow$ | Architecture |
| 5 | Chemistry (Chemical) | $\Longrightarrow$ | Cooking (Art) |
| | Temperature | $\longrightarrow$ | Heat |
| | Pressure | $\longrightarrow$ | Firepower |
| | Reactant Concentration | $\longrightarrow$ | Food Size |
| | Reactant | $\longrightarrow$ | Raw Material |
| | Product | $\longrightarrow$ | Dishes |

Table 7: Some data examples of SCAR. We give the system analogies ($\Longrightarrow$) with concept mappings ($\longrightarrow$).

the two concept mappings within the same system analogy to form a word analogy. For instance, in Fugure 1, we can obtain a word analogy, *i.e.*, *film is to retina as aperture is to pupil*. As reported in Table 6, our method exhibits a larger number of word analogies with bilingual language.

## D Details about Analogical Structure Abduction Task

We show the instructions combining backgrounds and the CoT prompting with explanations in Table 11 (I) and (II). One human-written instruction for the open analogical structure abduction task is shown in Table 8.

## E Embedding Similarity Method

We convert concepts in two systems into different embeddings with the following strategies and calculate cosine similarity between concepts:

1. Max-similarity algorithm, which establishes a mapping with the concept from System B that

*/* Task prompt */*
For two given systems, you are required to create an analogy by extracting concepts from the backgrounds of systems and matching the concepts in each system with one another in a one-to-one mapping.
*/* Data */*
**System A**: Camera
**System B**: Eye
**Background of System A**:
A camera is a device that captures visual images by...
**Background of System B**:
The eye is a remarkable organ that allows us...
*/* Question */*
**Question**: Please extract concepts from the backgrounds of systems and establish the mappings between concepts. The format should be a list:
(Concept1_SystemA, Concept1_SystemB),
(Concept2_SystemA, Concept2_SystemB), ...
*/* Answer */*
**Answer**: *(Film, Retina), (Diaphragm, Iris), (Aperture, Pupil), (Lens, Lens), (Black paint, Choroid)*

Table 8: An instruction template for LLMs in the open analogical structure abduction task. Generated texts by LLMs are *highlighted*.

exhibits the highest cosine similarity score. This method performs mapping *with replacement*, meaning that a concept from System B can be mapped to multiple concepts from System A;

2. Greedy algorithm (Zhang et al., 2000), which iteratively maps concepts exhibiting the highest cosine similarity in each step. In each round of iterations, we first calculate all cosine similarity scores between concepts in the two systems. Then, we map the concepts with the highest cosine similarity scores, and the concepts that have been mapped will not be considered in the next round of iterations. This strategy generates one-to-one mappings while not considering the overall optimality.

3. Kuhn-Munkres algorithm (Kuhn, 1955), a combinatorial optimization technique for solving one-to-one mapping problems. Given a matrix $\mathbf{C}$, with each $\mathbf{C}[\mathbf{i}, \mathbf{j}]$ representing the cost of matching vertex $i$ (a "worker") to vertex $j$ (a "job"), the objective is to find a minimal cost assignment of workers to jobs. Let $\mathbf{X}$ be a boolean matrix, with $\mathbf{C}[\mathbf{i}, \mathbf{j}] = 1$ if and only if row $i$ is assigned to column $j$. The optimal assignment cost is given by:

$$\min \sum_i \sum_j \mathbf{C}_{i,j} \mathbf{X}_{i,j}, \qquad (1)$$

where $\mathbf{X}$ is square, each row corresponds to exactly one column, and each column corresponds to exactly one row.

| I: Word Analogy Test | II: Relational Structure Identification (RSI) |
|---|---|
| */* Task prompt */*
Find the most analogous candidate answer that follows the relations in the query.
*/* Examples */*
Question: broom:dustpan
Choice:
A: lock:key
B: frame:lens
C: scarf:hat
D: toothbrush:cup
Please choose A, B, C or D.
Answer: D
*/* Test data */*
Question: admire:respect
Choice:
A: like:adore
B: oppress:exploit
C: spouse:husband and wife
D: relatives:neighbors
Please choose A, B, C or D.
Answer: *A* | */* Task prompt */*
What relationship is in the given analogy?
*/* Examples */*
Question: army:order:band::band leader
Choice:
A: govern
B: violate
C: obey
D: cooperate
Please choose A, B, C or D.
Answer: C
*/* Test data */*
Query: riverbank:bridge::floor:stairs
Choice:
A: separate
B: linked by
C: link
D: adjacent
Please choose A, B, C or D.
Answer: *B* |

Table 9: Prompt templates for LLMs in word analogy task and RSI task. Generated texts by LLMs are *highlighted*.

---

**I: Background Revision**

*/* Task prompt */*

Given a description of a system with a list of concepts related to the system, please generate a short introduction of the system according to the description and concepts within 500 words.

*/* Data */*

System: Biosphere

Description: The biosphere, also known as the ecosphere, is the worldwide sum of all ecosystems. It can also...

Concepts: biology, biodiversity, ecosystem

Introduction: *The biosphere refers to the sum of all ecosystems on Earth, including the interactions between living organisms and their environment...*

**II: Explanation Generation**

*/* Task prompt */*

Given two systems, each of which consists of some concepts, establishing mappings between the concepts can make the two systems analogous. Given mappings, please explain why such mappings can form an analogy. Hint: You can answer from a structural point of view.

*/* Data */*

System 1: Immune System

System 2: Army

Concepts in System 1:Immune Cells, Antibody, lymphoid tissue, regulatory organs

Concepts in System 2:Soldier, Weaponry, base, headquarters

Mappings: [Immune Cells,Soldier],[Antibody,Weaponry],[lymphoid tissue,base],[regulatory organs,headquarters]

Explanations:

*1. Immune cells correspond to soldiers: In the immune system, immune cells (such as white blood cells) are the soldiers responsible for identifying and eliminating pathogens. They can quickly recognize, attack, and clear invading pathogens. In the military, soldiers are the basic force for executing tasks and defending national security. In this mapping, immune cells and soldiers both represent the executors of their respective systems.*

*2. Antibodies correspond to weaponry: In the immune system, antibodies are specific proteins produced by immune cells. They can recognize and bind to pathogens, helping the immune system clear them. In the military, weaponry is the tool soldiers use to execute tasks. Antibodies and weaponry both represent specific tools used to accomplish tasks.*

*3. Lymphoid tissue corresponds to bases: In the immune system, lymphoid tissue (such as the spleen and lymph nodes) is where immune cells are produced, stored, and exchanged. In the military, bases are where soldiers live, train, and organize tasks. In this mapping, lymphoid tissue and bases are both support facilities for their respective systems.*

*4. Regulatory organs correspond to headquarters: In the immune system, regulatory organs (such as the thymus and bone marrow) are responsible for coordinating the production, development, and function of immune cells, ensuring the normal operation of the immune system. In the military, headquarters are responsible for organizing, commanding, and coordinating military actions. Regulatory organs and headquarters both represent the management and coordination organizations of their respective systems.*

Table 10: Prompt templates for background revision (generated by ChatGPT) and explanation generation (generated by GPT-4). Generated texts by ChatGPT and GPT-4 are *highlighted*.

| I: Instruction Template with Background |
|---|

*/* Task prompt */*

For two given systems, you are required to create an analogy by matching the concepts in each system with one another in a one-to-one mapping.

*/* Data */*

System A: Camera. **A camera is a device that captures images by exposing a photosensitive surface to light...**
System B: Eye. **The human eye is a complex optical instrument that allows us to perceive the world around...**
Concepts in System A: Film, Diaphragm, Aperture, Lens, Black paint
Concepts in System B: Iris, Choroid, Lens, Retina, Pupil
Please establish the analogies between the concepts.
The format should be a list: (Concept1_SystemA, Concept1_SystemB), (Concept2_SystemA, Concept2_SystemB), ...
Answer: *(Film, Retina), (Diaphragm, Iris), (Aperture, Pupil), (Lens, Lens), (Black paint, Choroid)*

| II: Instruction Template with Explanation (CoT) |
|---|

*/* Task prompt */*

For two given systems, you are required to create an analogy by matching the concepts in each system with one another in a one-to-one mapping.

*/* Example */*

System A: Immune System
System B: Army
Concepts in System A:Immune Cells, Antibody, lymphoid tissue, regulatory organs
Concepts in System B:base, Weaponry, headquarters, Soldier
Please establish the analogies between the concepts.
The format should be a list: (Concept1_SystemA, Concept1_SystemB), (Concept2_SystemA, Concept2_SystemB), ...
Answer: let's think step by step.
1. Immune cells correspond to soldiers: In the immune system, immune cells (such as white blood cells) are the soldiers responsible for identifying and eliminating pathogens. They can quickly recognize, attack, and clear invading pathogens. In the military, soldiers are the basic force for executing tasks and defending national security. In this mapping, immune cells and soldiers both represent the executors of their respective systems.
2. Antibodies correspond to weaponry: In the immune system, antibodies are specific proteins produced by immune cells. They can recognize and bind to pathogens, helping the immune system clear them. In the military, weaponry is the tool soldiers use to execute tasks. Antibodies and weaponry both represent specific tools used to accomplish tasks.
3. Lymphoid tissue corresponds to bases: In the immune system, lymphoid tissue (such as the spleen and lymph nodes) is where immune cells are produced, stored, and exchanged. In the military, bases are where soldiers live, train, and organize tasks. In this mapping, lymphoid tissue and bases are both support facilities for their respective systems.
4. Regulatory organs correspond to headquarters: In the immune system, regulatory organs (such as the thymus and bone marrow) are responsible for coordinating the production, development, and function of immune cells, ensuring the normal operation of the immune system. In the military, headquarters are responsible for organizing, commanding, and coordinating military actions. Regulatory organs and headquarters both represent the management and coordination organizations of their respective systems.
Therefore, the final answer is: (Immune Cells,Soldier), (Antibody,Weaponry), (lymphoid tissue,base), (regulatory organs,headquarters)

*/* Test Data */*

System A: Camera
System B: Eye
Concepts in System A: Film, Diaphragm, Aperture, Lens, Black paint
Concepts in System B: Iris, Choroid, Lens, Retina, Pupil
Please establish the analogies between the concepts.
The format should be a list: (Concept1_SystemA, Concept1_SystemB), (Concept2_SystemA, Concept2_SystemB), ...
Answer: let's think step by step.
*1. Film corresponds to Retina: ...*

Table 11: Instruction templates for LLMs in the structure mapping abduction task. Generated texts by LLMs are *highlighted*.