# OpenReview forum: "Beneath Surface Similarity: Large Language Models Make Reasonable Scientific Analogies after Structure Abduction"
_EMNLP/2023/Conference — EMNLP 2023 Findings_

### Official Review · Reviewer_1Lsr · 2023-08-02

**Soundness:** 4

**Excitement:**

4: Strong: This paper deepens the understanding of some phenomenon or lowers the barriers to an existing research direction.

**Paper Topic And Main Contributions:**

This paper is centrally interested in structured analogical reasoning in large language models. Existing work has focused on word analogies. This work suggests that results from that paradigm overestimate the overlap between neural models and human cognition. They introduce a task of analogical structure abduction that evaluates whether models can systemically associate the components of an analogy. The results suggest that models often miss the underlying relations, and thus, differ from humans. The main contributions are: i) a benchmark called ScAR and ii) a evaluation of LLMs on this task highlighting the need for future work to improve models.

**Questions For The Authors:**

A. Can you elaborate on the reasonableness of using the E-KAR dataset heavily in this work? The main concern is that it is a dataset taken from China’s Civil Service Examinations. The description provided in Appendix A suggests that “cultural knowledge” is part of the exam. Is it reasonable to expect that a model trained largely on English data and evaluated on English data will be able to answer these questions? I have in mind work like Santurkar et al. (2023). Whose Opinions Do Language Models Reflect. https://arxiv.org/abs/2303.17548., which demonstrate that models encode only some groups well.

B. Why was GPT-4 used so extensively in the creation of the dataset (e.g., for the explanation generation)? Wouldn’t higher quality data come from having the annotators do all of this (they are already reading through them after all)?

C. In the Ethics Statement it is stated that “as described in our paper, all annotators are compensated”, where is this stated in the body of the paper? It would be helpful if more details about the annotators (e.g., what are the 5 annotators backgrounds on lines 229-231) and their task were included (e.g., the instructions they were provided). This would facilitate both replication and extension of this work.

D. Technique for embeddings was unclear in section 2. It is stated that the method proposed by Ushio et al. (2021) was used (line 139). However, that paper used cosine similarity in evaluating embeddings, as far as I can tell. Moreover, it is stated that the winning answer candidate was determined with the marginal likelihood biased perplexity measure which uses probability of the token in context, not the embedding. Could this method be clarified in the paper?

MINOR:

E. In section 4, a few more models were evaluated. Why weren’t these models included in the other experiments?

F. The role of RSI in the paper is a bit unclear. Is it part of the new proposed benchmark as well? Or is it just used as an additional source of evidence that models fail to capture the underlying structural relations in an analogy (similar to how the ability of models to fully associate the set of concepts in the benchmark evidences they systematically understand the relations)?


**Reasons To Accept:**

The paper is well motivated and its aim clear. The benchmark will be of use to community and appears to address a gap in existing resources. The experiments are reasonably thorough and expose limitations with existing models, while suggesting ways they might be improved.

**Reasons To Reject:**

The problem identified by the paper is interesting and well articulated. There are a few methodological issues that weaken the rating (included under questions for the authors). Additionally, the linkage to human cognition (which appears to be a central aim of the paper) is a bit unclear. It is stated that the Structure Mapping Theory (SMT) has been proposed as a mechanism for analogical reasoning. Other times, it is stated that SMT suggests that new knowledge is gained by system level analogies. Put another way, there are two uses of SMT in the paper, (i) as the account for analogical reasoning broadly construed, and (ii) as the account for system level analogical reasoning (a more narrow form of analogy). If (ii) is the correct interpretation, then it’s possible models can do simple word analogies in a human-like fashion without the ability to do system level analogies. The lack of clarity around these possibilities makes it difficult to interpret what these results say about comparisons between models and humans.

Ultimately, the paper proposes a potentially useful benchmark for evaluating the underlying structural reasoning abilities of large-language models. However, some issues with the methodology and the underlying arguments motivating the conclusion depress my rating.


**Reproducibility:**

4: Could mostly reproduce the results, but there may be some variation because of sample variance or minor variations in their interpretation of the protocol or method.

**Reviewer Confidence:**

4: Quite sure. I tried to check the important points carefully. It's unlikely, though conceivable, that I missed something that should affect my ratings.

**Typos Grammar Style And Presentation Improvements:**

Figure 3d is used to argue that models are more robust in the CoT prompting scenario. The inclusion of error bars would facilitate more informed comparisons between the variation in a given condition and across conditions.

Consistent y-axis bounds in Figures 3, 5, and 6 would facilitate comparisons.

---

> ### Author Rebuttal · Authors · 2023-08-29
>
> Thank you for your thoughtful suggestions, we will make the following revisions to our paper:
>
> 1.**Transform Figure 3d into a bar chart accompanied by error bars** to better prove that providing an example, additional backgrounds, or utilizing CoT prompting can enhance the robustness of LLMs to the instruction design. Since the error bar is a specific confidence interval for the outcomes, here we report the **95% confidence interval** of the average system accuracy of LLMs with different methods for your reference:
> | Model  | 95% confidence interval| confidence interval width |
> |---------|:----:|:----:|
> | 0-shot     |[38.68, 43.54] |4.86|
> | 1-shot     |[43.13, 46.86] |3.73|
> | w/ Expl.   |[44.16, 47.08] |2.92|
> | w/ Backg   |[43.77, 46.73] |2.95|
>
> 2.Consistent y-axis bounds in Figures 3, 5, and 6 to facilitate comparisons.
>
> 3.Add detailed information about **annotations** to the Appendix to facilitate both replication and extension of this work (The information is shown in **Answer C**).
>
> ## Here is the answer to your concern
>
> > Weakness: The linkage to human cognition (which appears to be a central aim of the paper) is a bit unclear. There are two uses of SMT in the paper: (i) as the account for analogical reasoning broadly construed, and (ii) as the account for system-level analogical reasoning. If (ii) is the correct interpretation, then it’s possible models can do simple word analogies in a human-like fashion without the ability to do system-level analogies.
>
>
> Answer: Thank you for pointing out this confusion! We would like to clarify that both stated uses of SMT in the paper, (i) as the mechanism for broad analogical reasoning and (ii) as the mechanism for system-level analogical reasoning, are not contradictory. Instead, they demonstrate the versatility of SMT. The essence of SMT lies in identifying common relational structures among two "systems," where each system can be a central concept or situation with relevant concepts. Therefore, **system-level analogies can be generalized to broader ones**.
>
> According to SMT, analogical reasoning is based on identifying common relational structures between two systems. However, our RSI task shows that the word analogy test **can not** evaluate the ability of LMs to identify common relational structures between two systems. Therefore, even if a model can do simple word analogies in a human-like fashion and even achieves human-level performance on the word analogy test, it is **not guaranteed** to have human-level analogical reasoning ability.
>
>
> ## Here are the answers to your questions
> > Q(uestion)A: Can you elaborate on the reasonableness of using the E-KAR dataset heavily in this work? Is it reasonable to expect that a model trained largely on English data and evaluated on English data will be able to answer these questions?
>
> Answer A: As shown in Table 6, compared to other existing analogy benchmarks, E-KAR contains more data samples with **more diverse and challenging relations**, and it is the only benchmark that **provides relational explanations** for each pair of analogy. Additionally, as stated in the E-KAR's paper, E-KAR is **bilingual**, and the authors have **eliminated culture-specific samples** in the English split of E-KAR. Therefore, we primarily adopt E-KAR to construct the relational structure identification task dataset.
>
> > QB: Why was GPT-4 used so extensively in creating the dataset (e.g., for the explanation generation)? Wouldn’t higher-quality data come from having the annotators do all of this?
>
> Answer B: Providing human-written explanations could be costly, time-consuming and labor-intensive. Thus, we use GPT-4 to generate explanations and ask the annotators who created the dataset to revise the wrong explanations to **reduce human labor**, which has become a prevailing paradigm for data annotation nowadays (e.g., WANLI: Worker and AI Collaboration for Natural Language Inference Dataset Creation [Liu et al., EMNLP Findings 2022] and Reframing Human-AI Collaboration for Generating Free-Text Explanations [Wiegreffe et al., NAACL 2022]).
>
> > QC: In the Ethics Statement, it is stated that “as described in our paper, all annotators are compensated”, where is this stated in the body of the paper? It would be helpful if more details about the annotators (e.g., what are the 5 annotators' backgrounds on lines 229-231) and their tasks were included (e.g., the instructions they were provided).
>
> Answer C: Thank you for pointing out this confusion about annotation! Here's the detailed information about annotations for your reference, and we will add it in the revision:
>
> **Annotator Details**:
>
> We have recruited a team of five undergraduates with diverse academic backgrounds in Computer Science, History, Physics, Biology, and Chemistry. Among them, the student majoring in Computer Science has a minor in Economics, while the student majoring in History has minors in Creative Writing and Philosophy. We pay each annotator \\$8/h, exceeding the local minimum wage \\$5/h.
>
> **Guidelines For Annotators**:
>
> Instructions
>
> Thanks for participating in this HIT! Please spend some time reading this instruction and the example section to better understand our HIT! In this hit, you need first read the context of Structure Mapping Theory to learn how to identify potential analogies based on the relational structures of systems and then create analogies based on this theory as many as possible. **[Here, we introduced the structure mapping theory as in Section 2.1 of the paper. We omit it due to the length]** After reading the above context, we believe you have understood the SMT. Next, you need to create system analogies based on the relational structures of two systems. One data example is shown in Figure 1. You will complete 4 steps to create one system analogies.
> - Step 1: You need to write two systems that can form an analogy (e.g., Camera and Eye).
> - Step 2: You need to write relevant concepts of two systems as many as possible (e.g., Film, Diaphragm, Aperture, Lens, Black paint for Camera and Retina, Iris, Pupil, Lens, Choroid for Eye).
> - Step 3: You need to construct the mappings between concepts in two systems based on the relational structure abduction (e.g., aperture should be mapped to pupil).
> - Step 4: You need to remove the concepts that are not mapped.
>
> To facilitate a more efficient and effective annotation process, we provide you with scientific analogies sourced through online research **[We omit the URLs of these sources in Appendix B.1 due to the length]**. Notes: You can SEARCH GOOGLE to help you judge whether the mappings are reasonable, especially if you are unsure about the system.
>
> > QD: Technique for embeddings was unclear in section 2. It is stated that the method proposed by Ushio et al. (2021) was used (line 139). However, that paper used cosine similarity in evaluating embeddings, as far as I can tell. Moreover, it is stated that the winning answer candidate was determined with the marginal likelihood biased perplexity measure, which uses the probability of the token in context, not the embedding. Could this method be clarified in the paper?
>
> Answer D: We apologize for the confusion caused by this typo. We did use **cosine similarity in evaluating embeddings**. For the word embedding models, we follow the method proposed by Ushio et al. (2021) and represent word pairs by taking the difference between their InstructGPT embeddings (text-ada-embedding-002). Then, we choose the answer candidate with the highest cosine similarity to the query in terms of this vector difference.
>
> > QE: In section 4, a few more models were evaluated. Why weren’t these models included in the other experiments?
>
> Answer E: Thanks for your suggestion, and we will report the performance of all models in each experiment in the Appendix for reference in our next version. Due to limited space, we selected the **top three performing models** in the main results (Table 5) for further investigation (i.e., InstructGPT 003, ChatGPT and GPT-4).
>
> > QF: The role of RSI in the paper is a bit unclear. Is it part of the new proposed benchmark as well? Or is it just used as an additional source of evidence that models fail to capture the underlying structural relations in an analogy?
>
> Answer F: RSI **is not a part of** the new proposed benchmark. RSI is a task designed to have LLMs select the relation constituting the analogy, which is used to prove that the word analogy test **can not** evaluate the ability of LMs on identifying common relational structures between two systems.

---

### Official Review · Reviewer_SVGF · 2023-08-04

**Soundness:** 4

**Excitement:**

4: Strong: This paper deepens the understanding of some phenomenon or lowers the barriers to an existing research direction.

**Missing References:**

ACL 2023 has at least one paper on LLMs and analogical reasoning, which looks at abstract analogical reasoning: https://aclanthology.org/2023.acl-long.109/

This paper by Melanie Mitchell would be important to cite: https://arxiv.org/abs/2102.10717

**Paper Topic And Main Contributions:**

This paper is about analogical reasoning in language models supported by structure abduction. Its contributions are as follows:

1. The authors present a study on past analogical reasoning benchmarks to show that while LLMs can achieve high performance on simpler word analogy benchmarks, this is not always supported by an understanding of the underlying structure for the analogy (unlike humans). This motivates them to create a stronger evaluation for analogical reasoning that considers structure abduction jointly with making analogies.

2. The authors create a benchmark called Scientific Analogical Reasoning (ScAR) which provides 400 analogies across systems in 13 different scientific and other domains, including 1600 possible mappings between concepts within systems. They additionally collect background knowledge and explanations to support the analogies in the dataset, making it a possibly valuable and high-quality resource for studying analogy-making in LLMs.

3. The authors evaluate contemporary LLMs, both open-source and closed API-based, on ScAR under several prompting settings. Further, they perform analysis on the impact of various LLM-based aspects in the experiment and the domain. Lastly, they apply earlier word embedding-based methods for analogy as a reference, and perform a preliminary study on open analogical structure abduction, where concepts from the two analogous systems are not provided directly, and rather must be retrieved from a large list.

While the paper does not present new methods for tackling analogical reasoning with LLMs and perhaps lacks some consideration of the impact of LLM pre-training data on this problem, it still constitutes a substantial contribution that could be valuable to the NLP community.

**Questions For The Authors:**

**Question A:** To what extent can contamination of LLM pre-training data impact the observed abilities of LLMs in analogical reasoning? For example, I can Google about the aperture of a camera and pupil of a human eye and find plenty of web content talking about how they’re analogous. Does the LLM gain this capability just by memorizing the pre-training data?

I would be curious to see how thoroughly the system understands common systems or concepts across examples from ScAR - if the system can correctly answer the *aperture -> pupil* example, but can’t understand some other example about *aperture* that’s less common, this may suggest a lack of deep understanding of the concept of *aperture.* Figure 4 gives some hints about this possibility across domains, but doing such an analysis could also be a valuable addition to the work! *As a side note, this is one reason why Mitchell (2021) listed later in my review argues that we shouldn’t focus on natural language analogy problems, but I think this issue can also be explored by more thorough evaluations.*

**Reasons To Accept:**

**Strength A:** The paper compiles a (seemingly) high-quality benchmark dataset of interesting analogies across scientific and other domains, including consistent details about the underlying structure of analogies, and background knowledge and explanations for analogies. This resource could be useful to others in the community.

**Strength B:** The creation of the benchmark is justified by a preliminary study on how word analogy tasks do not actually represent analogical reasoning capabilities. I have not seen another work demonstrate this before, and this can be motivation for the community to look at analogical reasoning more holistically than prior work has.

**Strength C:** The analysis of benchmark results is thorough and fairly comprehensive, covering a lot of variations and details on the prompt engineering/LLM interface side, as well as including a thoughtful study on how analogies can be made across an open domain of concepts (which I believe is the hardest challenge in analogical reasoning). The authors also explore earlier representation/embedding-based approaches to analogy. It feels like the authors have considered a lot of questions in performing these experiments.

**Reasons To Reject:**

**Weakness A:** This paper doesn’t propose any new modeling techniques or methods to apply LLMs to analogy tasks. This causes the main benchmark results on the paper to not be very inspiring, as they don’t show much significant variation across existing models and prompting techniques that are explored.

**Weakness B:** The paper does not rigorously explore the impact of pre-training data on LLMs’ analogical reasoning. In natural domains like this that may be largely supported by memorization of training data, some analysis for this is important.

**Reproducibility:**

4: Could mostly reproduce the results, but there may be some variation because of sample variance or minor variations in their interpretation of the protocol or method.

**Reviewer Confidence:**

4: Quite sure. I tried to check the important points carefully. It's unlikely, though conceivable, that I missed something that should affect my ratings.

**Typos Grammar Style And Presentation Improvements:**

**Organization/presentation suggestions:**

L091: If accepted, you may want to use the extra space to break this paragraph out into a numbered list - may make it easier to read. There are some other paragraphs where you do this too, and I would recommend the same.

Section 3.4 presents a formal problem definition, which might make more sense to introduce before your dataset. This would help give the reader a foundation on what task you’re looking at, and then the data collection would be easier to understand and map each step to components of the task. I’m not totally sure - would leave it up to you.

I’m not sure if it’s fair to characterize Webb et al. (2022) as just looking at analogy generation with word analogies or simple sentence analogies as your Section 5 says around L529. This paper looks at abstract language-based analogy tasks and uses them to study how well LLMs can perform analogical reasoning. I might suggest adding a separate paragraph in Section 5 looking at abstract analogical reasoning rather than lumping it into the paragraph about analogies between situations. You may also think about organizing Section 5 based on how these different works focus on different parts of analogy-making, i.e., representation, mapping, and evaluation, and how your work contrasts from it.

**Formatting/typo corrections:**

L036, L046, L055, L060, L090, L110, L135, L344-348…: putting two sets of parentheses next to each other or nesting parentheses is improper format and a bit confusing to read. You can use \citealp in LaTeX to combine citations with other content in parentheses, or find another way to remove one set of parentheses.

L124: missing a space after period

Did you use ChatGPT or GPT-4 to revise background information about systems? The paper says ChatGPT, but Table 10 says GPT-4.

---

> ### Author Rebuttal · Authors · 2023-08-29
>
> Thank you for your thoughtful suggestions, we will make the following revisions to our paper:
> 1. Polish the paper to be more clear.
>     - Add the papers you mentioned in our paper.
>     - Break the paragraph in L091-L100 out into a numbered list.
>     - Put the formal problem definition into Section 3.1 before introducing our dataset.
>     - Use *\citealp* in LaTeX to combine citations with other content in parentheses.
>     - Correct the title of Table 10: *Prompt templates for background revision (generated by ChatGPT) and explanation generation (generated by GPT-4). Generated texts by ChatGPT and GPT-4 are highlighted.*
>
> 2. Reorganize Section 5 (Related Work) based on different parts of analogy-making, i.e., representation, mapping, and evaluation.
>
> 3. Add a separate paragraph about abstract analogical reasoning in Section 5, and refine the statement for better clarity on Webb's work: *"...Webb et al. (2022) examine the abstract language-based analogy task and evaluate analogical reasoning ability of LLMs on this task. Compared to their task, our task requires the intensive involvement of commonsense, encyclopedic, and cultural (e.g., idiom and historical) knowledge. "*
>
> ## Here are the answers to your concern
> > W(eakness)1: This paper doesn’t propose any new modeling techniques or methods to apply LLMs to analogy tasks. This causes the main benchmark results on the paper to not be very inspiring, as they don’t show much significant variation across existing models and prompting techniques that are explored.
>
> A(nswer)W(eakness)1: Thank you for your comments. Our work is not positioned as a method paper, but rather studies the problem of analogical reasoning for LLMs. However, to enrich the main benchmark results, we added **Llama 2 (13B and 70B)** as a baseline open-sourced LLM [1] and **self-consistency prompting** as a baseline for advanced prompting [2] on the English version of SCAR. For self-consistency prompting, we adopt two methods:
>
> (1) We set the temperature to 0.7, generate results **five times**, and take the majority vote to select the answer, which is denoted as w/ *Self-consistency (Temperature)*;
>
> (2) We use **five different instruction** templates and take the majority vote to select the answer, denoted as w/ *Self-consistency (Template)*.
>
> The results in the following table show that:
> 1. Llama 2 70B performs poorly, but self-consistency prompting can significantly improve performance.
> 2. Additionally, we observe performance improvements in both ChatGPT and GPT-4 through advanced prompting techniques like self-consistency prompting.
>
> [1] Touvron H, Martin L, Stone K, et al. Llama 2: Open foundation and fine-tuned chat models[J]. arXiv preprint arXiv:2307.09288, 2023.
>
> [2] Wang X, Wei J, Schuurmans D, et al. Self-consistency improves chain of thought reasoning in language models[J]. arXiv preprint arXiv:2203.11171, 2022.
>
> | Method  | Concept Acc.| System Acc. |
> |---------|:----:|:----:|
> | Llama 2 (13B)     |  36.78   |  19.50   |
> | w/ 1-shot     |  49.54   |  24.50   |
> | Llama 2 (70B)     |  52.38   |  32.50   |
> | w/ 1-shot     |   62.85  |  44.50   |
> | w/ Self-consistency (Temperature)  |  67.93   |  48.85   |
> | w/ Self-consistency (Template) |  58.14   |  46.50  |
> | ChatGPT     |  66.52   |  46.61   |
> | w/ Self-consistency (Temperature)  |  68.24   |  48.75  |
> | w/ Self-consistency (Template) |  68.68   |  53.50  |
> | GPT-4    |   73.28  |  58.50   |
> | w/ Self-consistency (Temperature)  |  77.96   |   67.00  |
> | w/ Self-consistency (Template) |  75.94   |  65.50  |
>
>
>
>
> > W2: The paper does not rigorously explore the impact of pre-training data on LLMs’ analogical reasoning.
>
> AW2: Please refer to Answer A for Question A
>
> ## Here is the answer to your question
> > Question A: To what extent can contamination of LLM pre-training data impact the observed abilities of LLMs in analogical reasoning?
>
> Answer A: Thank you for this suggestion. We randomly sample 10% (about 35M samples) of the C4 dataset, which is from Common Crawl's web corpus, a commonly used source for pre-training LLMs, to observe the **contextual (co-occurrence) associations** between the concepts of two systems in the pre-training data.
> We aggregate the number of articles in which two systems' concepts co-occurred. In the C4 dataset, each sample corresponds to an article. If the concepts of two systems are in a single article, it might indicate a high co-occurrence between the concepts. The results are listed below.
> System A -> System B | System A in article| System B in article| # Co-occurred / # Total |
> |---------|:----:|:----:|:----:|
> Camera -> Eye | ~1M | ~2M | ~17K/35M   |
> Cell -> Factory | ~4K | ~238K |  27/35M   |
> Computer Systems -> Urban | ~483K | ~1M |  541/35M   |
> Chemistry -> Cooking | ~82K | ~853K |  93/35M   |
>
> We find that the concepts of two systems from different domains rarely co-occurred. Yet, LLMs can still establish the mappings between concepts to form analogies and recover reasonable explanations of relational structures. The low co-occurrence between concepts suggests that LLMs may **not rely on the pre-training data's co-occurrence associations** for analogical reasoning. Therefore, we think that the observed abilities are not severely contaminated by the pre-training data. Instead, pre-training teaches LLMs **a wealth of background knowledge**, enabling them to identify and infer relational structures behind analogical reasoning.

---

### Official Review · Reviewer_6uTV · 2023-08-05

**Soundness:** 2

**Excitement:**

2: Mediocre: This paper makes marginal contributions (vs non-contemporaneous work), so I would rather not see it in the conference.

**Justification For Ethical Concerns:**

The annotation was done by human annotators at "local" minimum wage.  It is not said where those annotators are from or under what conditions they are performing the annotation.

This concern was addressed sufficiently by the authors.   They used undergraduates who were trained in this particular task.

**Paper Topic And Main Contributions:**

The paper proposes to move beyond word analogies for LLMs and to enable "analogical structure abduction".  To this end a new data set is constructed based on the efforts of human annotators paid a local minimum wage, wherever local may be and however much that works out to be.  The results are mixed on a closed list of analogies, quite bad on an open list.  The main contributions are a move towards going beyond word analogies and the creation of a new data set.

**Questions For The Authors:**

It is not clear to me how the classic "queen is to woman as king is man" is really an instance of analogy since basically only the similarity of two vectors to one another  is being compared: how is that really analogical reasoning?   Do the authors equate semantic similarity with analogy?  This point needs to be brought out more strongly in the paper.

**Reasons To Accept:**

The topic of the paper is interesting given that, as the authors point out, analogy is a very basic human skill that is applied across many different domains. It is as such interesting to extend LLMs capabilities to include analogical reasoning.

**Reasons To Reject:**

Within linguistics, language change based on analogy is pervasive as is language acquisition based on rule formation via analogy.  Analogy is also a powerful tool for story-telling, writing and argumentation.  Yet the authors chose none of these domains for their experiments.  Rather, they construct an artificial task of a slightly more complex relationship between words that looking at word similarity in that the relationship involves pairs of two words rather than a single pair of words.  It is not clear how this really reflects the human process of analogy or how it is more generally of interest beyond this rather artificially constructed sets of relations between pairs of words.  This is also presumably why the attempts at having LLMs work with the open list of "analogies" do so badly: the task is artificially defined and does not tap into higher principles/generalizations that the LLMs might have learned.

**Reproducibility:**

4: Could mostly reproduce the results, but there may be some variation because of sample variance or minor variations in their interpretation of the protocol or method.

**Reviewer Confidence:**

3: Pretty sure, but there's a chance I missed something. Although I have a good feel for this area in general, I did not carefully check the paper's details, e.g., the math, experimental design, or novelty.

---

> ### Author Rebuttal · Authors · 2023-08-29
>
> Thank you for your valuable feedback.
> ## Here are the answers to your concern
> In order to address your concern more clearly, we divide your concern into several parts and respond to each of them separately.
> > W(eakness)P(art)1: Within linguistics, language change based on analogy is pervasive, as is language acquisition based on rule formation via analogy. Analogy is also a powerful tool for story-telling, writing and argumentation. However, the authors chose none of these domains for their experiments.
>
> A(nswer)W(eakness)P(art)1: We agree that analogy is useful in applications like story-telling, writing, and argumentation. In this work, we focus on studying the fundamental structure of analogical reasoning beneath these applications, according to the Structure Mapping Theory (SMT) of analogical reasoning in humans.
> For example, authors often use analogies such as "love is a journey" in novels, which could be a guide to the storyline. This analogy **essentially establishes mappings** between the concepts of "love" and "journey", where "lover" maps "traveler", "relationship" maps "road", "stage" maps "milestone", "difficulty" maps "obstacle" and "happiness or separation" maps "destination".
> The analogical structure abduction task introduced in this paper examines the process behind such analogy-making process. This task can **simulate this mapping process and thus can explain the rationale behind the analogy-making in story-telling, writing, and argumentation**. Moreover, our benchmark includes analogies from various domains such as Philosophy, Art, and Literature, which can be a valuable resource to understand and improve LMs' abilities in writing with analogy-making.
>
> > WP2: Rather, they construct an artificial task of a slightly more complex relationship between words that looks at word similarity in that the relationship involves pairs of two words rather than a single pair of words. It is not clear how this really reflects the human process of analogy.
>
> AWP2: Our task is a simplification of modeling what actually happens beneath analogical reasoning by SMT, i.e., **identifying common relational structures** between two systems. An example (i.e., camera and eye) is illustrated in Figure 1 in our paper. Based on the relational structure, the aperture should be mapped to the pupil since both are channels for light to enter, while black paint maps the choroid since both absorb light to prevent it from bouncing and reflecting. Thus, our task is a **faithful simulation of this process**.
> Furthermore, as shown in Figure 5 in our paper, relying solely on word similarities can not achieve satisfactory performance on our task. Therefore, our tasks need **a comprehensive understanding of systems and relevant concepts and the relational structure among them** rather than only focusing on pairs of two words and looking at word similarity.
>
> > WP3:  It is not clear how it is more generally of interest beyond this rather artificially constructed set of relations between pairs of words. This is also presumably why the attempts at having LLMs work with the open list of "analogies" do so badly: the task is artificially defined and does not tap into higher principles/generalizations that the LLMs might have learned.
>
> AWP3: We believe this task (analogical structure abduction) is a foundation for the **development, explanation, and evaluation** of more complex analogies that underlie various creative and cognitive processes. For example, the analogy "ideas are seeds" forms the basis of concepts and storylines, when extended to real applications like novel writing. As "seeds" need "nurturing" to "grow", "ideas" require "development" to "reach their full potential". The process of basic word-level analogy serves as the stepping stone to **creating and explaining** more complex analogical reasoning in real-world applications.
> By simplifying the complex analogy-making process to the fundamental analogy-making with words and concepts, we can evaluate the LLMs' ability for analogical reasoning in a **more focused and effective way**. This **disentangles confounding factors for evaluating analogical reasoning in real-world tasks**, such as narrative composition or rhetorical complexity.
> Therefore, our task forms **a part of a whole framework for assessing and bolstering the ability of LLMs to generalize and apply analogical reasoning** in more intricate, real-world situations.
>
> ## Here is the answer to your question
>
> > Question: It is not clear to me how the classic "queen is to woman as king is man" is really an instance of analogy since basically only the similarity of two vectors to one another is being compared: how is that really analogical reasoning? Do the authors equate semantic similarity with analogy? This point needs to be brought out more strongly in the paper.
>
> Answer: In the example "queen is to woman as king is man", the analogy is **not merely identifying the semantic similarity** between them. Instead, it recognizes the shared relation "gender of a ruler".
> In cognitive science, analogy can be used to convey that two situations or systems share a relational structure despite differences in the objects that make up the domains. This means that common relations (**structural similarity**) are essential to analogy, not the similarity of objects.
> In our paper, we argue that the word analogy test (*A is to B as C is to D*) does not explicitly evaluate **structural similarity** between two systems and thus propose an analogical structure abduction task to evaluate analogical reasoning of LLMs which are better aligned with human cognition.
>
> ## Here is the answer to the ethical concern
> > Ethical Concern: The annotation was done by human annotators at "local" minimum wage. It is not said where those annotators are from or under what conditions they perform the annotation.
>
> Answer: We recruited a team of five undergraduate students with different academic backgrounds to annotate this benchmark. The annotators are provided with guidelines of SMT to learn about identifying potential analogies based on the relational structures of systems. We pay each annotator $8/h, exceeding the local minimum wage $5/h. Please refer to **Answer C for Reviewer 1Lsr** for detailed annotation information.

---

### Meta-Review · Area_Chair_77sm · 2023-09-15

**Recommendation:** 4

**Metareview:**

The paper explores analogical reasoning in large language models (LLMs), moving beyond word analogies to "analogical structure abduction". It introduces a benchmark, Scientific Analogical Reasoning (ScAR), with 400 analogies across 13 domains and 1600 concept mappings, supplemented with background knowledge. The study reveals LLMs, while adept at word analogies, often miss deeper structural connections, unlike humans. Using ScAR, LLMs show varied results, particularly when structures aren't directly provided. The paper's key contributions are the ScAR benchmark and highlighting the LLM-human cognition gap in analogy. It paves the way for future research in this area.

The paper's blend of a relevant topic, novel insights, a high-quality dataset, and comprehensive analysis makes it a worthy contribution to the field. In particular:
a) The authors have meticulously compiled a benchmark dataset that encompasses a diverse range of analogies spanning scientific and other domains. This dataset is enriched with consistent details about the structural essence of analogies and is supplemented with background knowledge and explanations. Such a comprehensive resource stands to benefit the broader community.
b) A remarkable feature of the paper is its preliminary study highlighting the inadequacies of word analogy tasks in truly representing analogical reasoning capabilities. This perspective, not widely explored in previous works, offers a fresh lens to view analogical reasoning, urging the community to adopt a more holistic approach.

However, the main weaknesses of the contribution are: paper's lack of novel techniques (despite the aforementioned novel insights), insufficient exploration of pre-training data, ambiguity in its linkage to human cognition, and methodological shortcomings.

---

### Decision · Program_Chairs · 2023-10-07

**Decision:**

Accept-Findings

**Comment:**

The paper explores analogical reasoning in large language models (LLMs), moving beyond word analogies to "analogical structure abduction". It introduces a benchmark, Scientific Analogical Reasoning (ScAR), with 400 analogies across 13 domains and 1600 concept mappings, supplemented with background knowledge. The study reveals LLMs, while adept at word analogies, often miss deeper structural connections, unlike humans. Using ScAR, LLMs show varied results, particularly when structures aren't directly provided. The paper's key contributions are the ScAR benchmark and highlighting the LLM-human cognition gap in analogy. It paves the way for future research in this area.

The paper's blend of a relevant topic, novel insights, a high-quality dataset, and comprehensive analysis makes it a worthy contribution to the field. In particular:
a) The authors have meticulously compiled a benchmark dataset that encompasses a diverse range of analogies spanning scientific and other domains. This dataset is enriched with consistent details about the structural essence of analogies and is supplemented with background knowledge and explanations. Such a comprehensive resource stands to benefit the broader community.
b) A remarkable feature of the paper is its preliminary study highlighting the inadequacies of word analogy tasks in truly representing analogical reasoning capabilities. This perspective, not widely explored in previous works, offers a fresh lens to view analogical reasoning, urging the community to adopt a more holistic approach.

However, the main weaknesses of the contribution are: paper's lack of novel techniques (despite the aforementioned novel insights), insufficient exploration of pre-training data, ambiguity in its linkage to human cognition, and methodological shortcomings.